# Clinical and Obstetric Risk Factors for Postnatal Depression in HIV Positive Women: A Cross Sectional Study in Health Facilities in Rural KwaZulu-Natal

**DOI:** 10.3390/ijerph17228425

**Published:** 2020-11-14

**Authors:** Nontokozo Lilian Mbatha, Kebogile Elizabeth Mokwena, Sphiwe Madiba

**Affiliations:** Department of Public Health, School of Health Care Sciences, Sefako Makgatho Health Sciences University, Pretoria 0001, South Africa; lilian.mbatha@gmail.com (N.L.M.); Sphiwe.madiba@smu.ac.za (S.M.)

**Keywords:** South Africa, rural, HIV, maternal health, postnatal depression, PMTCT, mental health

## Abstract

Postnatal depression (PND) remains underdiagnosed and undertreated in different socio-economic backgrounds in South Africa. This study determined the prevalence of and clinical and obstetric risk factors for PND symptoms among HIV positive women in health facilities in a rural health district in South Africa. The Edinburgh Postnatal Depression Scale was used to measure PND from 386 women who had delivered a live infant. More than half (58.5%) tested HIV positive during the current pregnancy. The prevalence of PND symptoms was 42.5%. Logistic regression analysis yielded significant associations between clinical and obstetric variables of pre-term baby (*p*-value < 0.01), baby health status *p*-value < 0.01), baby hospitalization, (*p*-value < 0.01), and knowing the baby’s HIV status (*p*-value = 0.047). Maternal variables associated with PND were level of education (*p*-value < 0.01), monthly income (*p*-value < 0.01), and source of income (*p*-value = 0.05). At multivariate analysis, none of the clinical and obstetrical risk factors were independently associated with the PND. The high prevalence of PND symptoms underscore the need to integrate routine screening for PND in prevention of mother to child transmission of HIV programmes to enable early diagnosing and treatment of PND.

## 1. Introduction

Globally, maternal mental health problems continue to present a public health challenge. The mental ill-health of women of childbearing age and related factors are increasingly affecting the women’s quality of life [1]. The prevalence of depression in women in low and middle-income countries (LMICs) is estimated at 15–57% [2,3,4]. Postnatal depression (PND), in particular, is a maternal health phenomenon which occurs frequently among women who have recently given birth [5]. A systematic review reported a prevalence of PND of 19.8% from studies conducted in LMICs [5]. PND has long-term negative physical and psychological consequences for both the mother and the child. PND can lead to chronic depression and disruptions of family and marital relationships.

In LMICs, maternal depression is positively associated with impaired child growth [6]. PND adversely affects the socio-emotional, cognitive, behavioral, and psychomotor development of children. In addition, PND increase the risk of perinatal HIV transmission and poor physical health [7]. Literature further suggests that PND increases the risk of child mortality. A study conducted in Ghana, for example, reported that infants born to women with PND had a threefold increased risk of mortality by six months [8]. Moreover, the long-term consequences are not only limited to infancy, but extend through to toddlerhood and preschool age and can persist into late adolescence [9,10,11].

Depression is the most common neuropsychiatric complication in people living with HIV (PLHIV) with a prevalence estimated to be as high as 42% on average [12], and HIV positive women are particularly vulnerable to depression, particularly in sub-Saharan Africa [4,13]. In a study conducted in South Africa among PLHIV, the results indicated that 22% of PLHIV were severely depressed [14]. HIV positive women are particularly vulnerable to depression and this is particularly so in sub-Saharan Africa [4,13]. Amongst PLHIV, particularly in LMICs, depression is known to hasten HIV disease progression [15].

Maternal depression is among the most prevalent comorbidities among People Living With HIV/AIDS than the general population globally [12,16]. Research indicates that pregnant women living with HIV (Women Living with HIV) have increased risk of postpartum depression symptoms because of numerous stressors, including concern for the wellbeing of their infants [14,17,18,19]. Learning about their HIV infection during pregnancy adversely impacts on the mental health of women [20]. Studies conducted in sub-Saharan Africa found an increased risk of PND among women who discovered their HIV serostatus during pregnancy [18,21,22]. Furthermore, the pressures of having to cope with the diagnosis of HIV, the uncertainty about the baby, and all the associated potential problems places WLHIV in an even more disadvantaged position than those who are HIV negative [23,24,25].

Given this fact, depression then presents a significant challenge to WLHIV, especially in the perinatal period [18,26]. The prevalence of PND among WLHIV in SSA is high. Kaida et al. [3] reported prevalence of 39% in Uganda, whereas Dow et al. [17] reported prevalence of 33.5% in Malawi. In South Africa, the prevalence of PND among WLHIV ranges between 42.2–49.3% [13,19,27,28]. Recent studies in South Africa found that almost half of WLHIV reported high prevalence of PND. Mokwena and Shiba [28] and Tuthill et al. [29] reported prevalence of PND of 49.3% and 48.4% in Gauteng and KZN provinces, respectively.

Factors associated with PND are multifactorial resulting from the interaction of several risk factors including biological, psychological, and social. Maternal history of depression, immune system processes and adverse life events have all been independently associated with PND [30]. Maternal and obstetrical factors such as unplanned pregnancies, maternal age, and low level of education, parity, delivery via caesarean sections, pregnancy and delivery complications, and socio-economic status increase the risk of PND for HIV positive and negative women [31,32,33,34]. Baby related factors associated with increased risk of PND include the age of the baby, health status, and prematurity [35].

KwaZulu-Natal has the highest antenatal HIV sero-prevalence at 44.4%, with a reported 2% increase each year [36]. The high antenatal HIV sero-prevalence indicates that a high proportion of the women discover their HIV serostatus during pregnancy, which increases the risk of psychological distress, including antenatal and postnatal depression [37]. For an HIV positive woman, PND has a negative impact on the prevention of mother-to-child transmission of HIV infection. The internalized stigma related to an HIV diagnosis increases the risk of PND which may negatively affect the uptake of the antenatal care, obstetrics medication, and ART adherence [14,17,18,19].

However, PND remains underdiagnosed and undertreated among WLHIV in different socio-economic backgrounds in South Africa. This study determined the prevalence of postnatal depression symptoms among HIV positive women and identified clinical and obstetric factors associated with PND. It is important that risk factors predisposing to PND are well documented [38] to meet the needs of WLHIV within the antenatal and postnatal period.

## 2. Materials and Methods

### 2.1. Study Design and Setting

The study was a descriptive cross-sectional survey aimed at determining the prevalence of postnatal depression and related factors among HIV positive women. The study population consisted of HIV positive mothers who had delivered a live infant within 1–12 weeks of the time of study and were attending primary health care (PHC) facilities for postnatal care and Prevention of Mother To Child Transmission services. The mothers were recruited from PHC facilities in the uMhlathuze Municipality in the King Cetshwayo district with a population of about 352,003 in 2017. There are 13 PHC facilities, with a PHC utilization of 3.2 visits per person per year in 2017 and a postnatal clinic visit rate of 56.4% in six days [39]. A sample size of 380 was determined using the online Raosoft Sample Size Calculator at a confidence level of 95% and 5% error margin [40].

#### 2.1.1. Measures

A questionnaire and a screening tool—the Edinburgh Postnatal Depression Scale (EPDS)—were used, and both were translated and available in either IsiZulu or English which are the most frequently spoken languages in the district.

##### Questionnaire

A structured questionnaire was used to collect sociodemographic, baby and obstetric and HIV-related information. Sociodemographic items included maternal age, marital status, place of residence, religion, level of education, employment status, monthly income, and source of income. The baby and obstetric variables included parity, delivery method, gestation period, planned pregnancy, baby sex, and baby sex preference, history of baby hospitalization, feeding method, and baby HIV status. HIV related variables included time since HIV diagnosis, disclosure, and partner HIV status.

##### Edinburgh Postnatal Depression Scale

The 10-item Edinburgh Postnatal Depression Scale (EPDS), a validated screening tool, was used to screen the prevalence of PND symptoms. This screening tool has been used in several studies in Africa and in South Africa [4,28,41]. The EPDS consists of 10 self-reporting questions which identify different clinical depression symptoms such as anxiety, sleep disturbance, a feeling of guilt, helplessness, tearfulness, and a lack of motivation. Each question presents four answer choices scored on an ordinal scale of 0 to 3. The responses are summed to a possible total score of 0 to 30 [42]. A greater score indicates the likelihood of PND. Although the EPDS score threshold varies, higher cut off scores yield greater specificity [43]. According to Lawrie et al. [41] a score of 11 or 12 identified 100% of women with major depression and 70.6% of those with minor depression. The sensitivity was 80%, specificity 76.6%, a positive predictive value 52.6% and a negative predictive value 92.2%. The EPDS is identified as a reliable and valid measure to screen for PND in Africa. For this study, a threshold of 12 and above was used to determine the likelihood of postnatal depression. Pretesting of the questionnaire and the screening tool was done prior to data collection at one of the local clinics in the district. The questionnaire was administered to 15 women. After analyzing data from the pilot study, unclear questions were rephrased to ensure that the responses would be meaningful and valid.

### 2.2. Data Collection

A convenient sampling technique was used. All mothers with a live infant within 1–12 weeks visiting the PHC for postnatal services or routine immunization were recruited to participate. Prior to data collection, the researcher assistant was trained by the researcher on administering of the tool, and both the lead author and a trained research assistant collected data over a period of four months. The postnatal mothers participated in the study as they came for their postnatal services at their local clinics. Informed consent was obtained before completing the questionnaire. The researcher used all means available to ensure the participants’ privacy during the interview or the completion of the questionnaire. Hence the completion of the questionnaire and the screening tool were completed individually, and the completion of the questionnaire took place away from the waiting queue.

### 2.3. Data Analysis

To ensure the accuracy of the analysis, the data spreadsheet was imported to an interactive data analysis statistical package, STATA 13.0, for analysis (Stata Corp, College Station, TX, USA). Descriptive statistics included summary and frequency count for all sociodemographic, baby, and obstetric and HIV related responses. The paired Student *t*-test was used to compare the means of the continuous variables. Bivariate analysis was used to investigate the bivariate associations between the sociodemographic, baby, and obstetric variables and PND. Multiple logistic regression was used to determine independent risk factors associated with PND and also to estimate odds ratios and 95% confidence intervals for the risk factors. Statistical significance was defined as *p* ≤ 0.05. A score of 12 on the EPDS was used as a threshold for determining the likelihood of PND.

### 2.4. Ethical Considerations

Ethical clearance was obtained from the Sefako Makgatho Health Sciences University Research Ethics Committee (SMUREC/H/291/2017: PG) and permission was granted by the KwaZulu Natal Health Research and Knowledge Management Committee.

## 3. Results

### 3.1. Description of the Sample

The sample consisted of 386 HIV positive mothers with a live baby aged between 1 and 12 weeks. As illustrated in Table 1, the age of the participants ranged from 16 to 42 years with a mean age of 29 years (SD = 5.8 years). The majority (330/85.5%) were single, 316 (81.9%) lived in rural areas, 110 (28.5%) had not completed grade 12, 304 (78.8%) were unemployed, 120 (64.9%) earned less than R2000 (equivalent to $140.76) per month, and 183 (53.1%) reported the child support grant (R420 equivalent to $29.56) as their main source of income. More than two-thirds (272/70.5%) were multiparous, and only a third (124/32.7%) had planned the last pregnancy. Almost half (189/49.2%) of the women were diagnosed with HIV within the past year and, of those, 223 (58.5%) were diagnosed with HIV during the current pregnancy, 76 (19.8%) had not disclosed their HIV serostatus to family members, 125 (33.5%) reported unknown partner’s status, and 60 (16.1%) had HIV negative partners.

Over one-third (*n* = 147, 38%) of the babies were aged between 5–8 weeks, with a mean age of 6.4 weeks (SD = 3.5). Half (*n* = 194, 50%) of the babies were girls, 67 (17.4%) were born premature, 255 (66.1%) were born via normal vaginal delivery, and 251 (65.3%) were born in hospitals. A few (59/15.7%) babies had a history of hospitalization, 356 (92.2%) were reported to be in good health, 222 (57%) were breastfeeding, and 128 (33.2%) of the babies’ gender was not the parents’ preferred gender.

### 3.2. Maternal Factors Associated with PND Symptoms

An EPDS score cut of ≥12 was used as a threshold for determining the likelihood of PND. The EPDS scores ranged from 0 to 30 and the mean EPDS score of the participants was 10.7, (SD = 6.1). Of the 386 mothers, 164 (42.5%) had EPDS scores of 12 or greater, and these mothers screened positive for PND and were considered at risk for PND. Among participants who screened positive for PND (Table 2) PND symptoms were reported by a greater proportion of participants who knew the partner’s HIV status than those who did not (43.6 vs. 30%), those who knew their baby’s HIV status than those who did not (47.5 vs. 35.3%), those who were diagnosed with HIV during pregnancy than those diagnosed before pregnancy (55.6 vs. 44.4%), those who were single than those who were married (43.3 vs. 37.5%), and those who were aged above 35 years than those less than 35 years (45 vs. 41%).

In bivariate logistic regression analyses, maternal factors that were found to be statistically significantly associated with PND symptoms were; monthly income, source of income, level of education, and knowing the baby’s HIV status. Mothers with a monthly income of <R2000 were twice more likely to report PND symptoms (OR = 2.3, CI: 1.45–3.54, *p*-value < 0.01) than those with a monthly income of >R2000. Those whose source of income was the child support grant were 1.5 times more likely to report PND symptoms (OR = 1.5, CI: 1.10–2.36, *p*-value = 0.05) than those reporting other sources of income. Those who had completed grade 12 had half the risk of reporting PND symptoms (OR = 0.5, CI: 0.32–0.89, *p*-value = 0.003) than those who did not complete grade 12. Mothers who knew the HIV status of their babies had a 70% less chance of reporting PND symptoms (OR = 0.7, CI: 0.58–0.99, *p*-value = 0.047) than those awaiting the HIV test results of their babies.

### 3.3. Baby and Obstetric Factors Associated with PND Symptoms

Among those that reported most PND symptoms were women who had delivered via C-section than those who had normal vaginal delivered (45 vs. 41.2%), those who delivered at home than those who delivered at the hospital (66.7 vs. 43.8%), those who reported that the pregnancy was unplanned than those who planned the pregnancy (44.3 vs. 39.5%), those who had premature babies than those who had full term babies (67.2 vs. 37.6%), those who reported a history of baby hospitalization than those with no history of baby hospitalization (64.4 vs. 38.5%), those who reported their baby’s ill health than those with healthy babies (73.3 vs. 39.9%), those who were mix-feeding than those who were exclusively breastfeeding (67.7 vs. 42 and 43.2%), and those who did not have a baby sex preference than those with a preferred baby sex (49.1 vs. 39.5%).

At bivariate logistic regression analysis, baby and obstetric factors associated with PND symptoms were prematurity, baby history of hospitalization, and baby’s ill health. The results showed that mothers who had a premature birth were almost four times more likely to report PND symptoms (OR = 3.4, CI: 1.86,6.32, *p* = 0.00) than mothers who had a full-term birth. Mothers who reported history of hospitalization of the baby were three times more likely to report PND symptoms (OR = 2.9, CI: 1.62–5.26, *p* = 0.00) than mothers with no history of hospitalization. Mothers with sick babies were four times more likely to report PND symptoms (OR = 4.1, CI: 1.71–11.03, *p* = 0.00) than those with generally healthy baby (Table 3).

### 3.4. Multivariate Analysis of Factors Associated with PND Symptoms

A multiple logistic regression analysis was conducted to identify independent variables that are statistically significant associated with PND (Table 4). The results showed that mothers with an income <R2000 are twice more likely to report PND symptoms (OR = 2.3, CI: 1.17 4.35, *p*-value = 0.015) than those earning ≥R2000. The source of income was also independently associated with PND symptoms. Mothers with a social grant as the only source of income were almost three times more likely to report PND symptoms (OR = 2.8, CI: 1.15–6.88, *p*-value = 0.023) than those who had partner or family support (Table 4).

## 4. Discussion

The paper reports on the clinical, baby, and obstetric factors associated with PND among HIV positive women attending PHC in a rural district in KwaZulu-Natal. The study population was characterized by a high unemployment rate, low income of less than R2000 ($140) per month, and being dependent upon the government for health services. Most of the women were young and dependent on child support grants or family/relative support as the main source of income.

The research results indicated high prevalence of PND symptoms (42.5%). The prevalence of PND of 42.5% as measured by the EPDS with a cut-off point ≥12 is slightly lower than the prevalence found in a previous study in Mpumalanga Province, South Africa. Peltzer and Shikwane [19] reported a prevalence of 45.1% in similar study populations of HIV positive women in rural settings. A study conducted in a rural district in Cape Town reported a much higher prevalence of PND of 50.3% among positive and negative women [44]. However, the current study prevalence falls within the range of prior prevalence estimates reported in sub-Saharan Africa and South Africa (42.2–49.3%), such as a prevalence of depressive symptoms of 43% reported by Kakyo et al. [45] in a rural district in Uganda. The prevalence is also lower than the 52.5% reported from a study conducted in India among women on ART [46].

In this study, at bivariate logistic regression, preterm delivery, history of baby hospitalization, baby health status, knowing baby’s HIV status, monthly income, and source of income were significantly associated with PND. The results showed that mothers of pre-term babies were thrice more likely to report PND symptoms, compared to mothers who had full term babies. In contrast to previous research conducted with similar population [19], pre-term delivery was not associated with PND. The history of hospitalization of the baby was associated with an increased risk of PND, this is consistent with other studies which reported that illness of the infant and the admission of an infant to hospital contributed to the increase risk for PND among HIV positive mothers [17,47,48,49]. Similar to the current study, a systematic review of studies conducted in low- and middle-income countries found that the prevalence of PND was higher among mothers whose infants were ill than among those whose infants were well. Mothers may feel distressed because their infants are sick, thus increasing the risk of PND [5].

This study found that mothers who did not know their baby’s HIV status were at risk of PND, and a quarter (25.3%) of the mothers in this study were awaiting their baby’s HIV test results at the time of data collection. According to Dow et al. [17], there is likelihood that a reaction to the news of the infant’s HIV positive status may result in higher prevalence of PND symptoms. Similarly, mothers may feel distressed when they await the infant’s status results, which poses as a risk factor for PND. Furthermore, HIV-infected women may experience anxiety over the health of their infected infant or the ongoing risk of transmission of HIV to the infant if the child is uninfected [17].

The risk of PND was reduced for women who completed the 12th grade, which is consistent with those reported in other studies which has established a low prevalence of PND associated with a high level of education among positive and negative women in LMICs [5,26,50,51]. The study found that monthly income and the source of income are independent predictors of PND symptoms, which confirms the results of other studies which found an association between economic deprivation and PND. In a study done in KwaZulu-Natal, poverty and HIV and AIDS were identified as drivers of distress [52], and in a study conducted in India, poverty was an independent predictor of PND [53]. Low income was identified as a predictor of PND in LMICs [2,3,5,54].

The findings of this study showed a number of similarities and dissimilarities with other studies conducted in LMICs. Among the similarities are sociodemographic factors, such as marital status and place of living, were not associated with PND [2,3,54] and that obstetrical factors, such parity and unplanned pregnancy, were not associated with PND [53]. The findings that the sex of the baby and gender preferences were not associated with PND contrasted with the findings of Shivalli and Gururaj [53], who found an association between these baby factors and PND. These findings suggest that the association between demographic variables and PND are not universally consistent but case and setting specific.

Furthermore, there was no association between non-disclosure of HIV positive status and the development of PND, even though a quarter of the mothers who reported PND symptoms had not disclosed their status to any family member and did not know their partner’s HIV status. Sarna et al. [46] suggest that non-disclosure of HIV status to family members due to stigma and concerns about having a HIV negative child may predispose women to PND.

The main limitation of the study was that the population consisted of only Black postnatal women, which does not enable the generalization of the findings to other racial groups. The EPNDS is a screening and not a diagnostic tool and there were no follow-ups to confirm a clinical diagnosis.

## 5. Conclusions

The high HIV prevalence in South Africa may result in a tendency to focus on the physical health outcomes of the mother and may inadvertently downplay the importance of paying attention to the mental health aspect, and specifically PND, which has a significant impact on both the physical and mental health of both the mother and the baby. Services for the management of HIV in primary health facilities typically include some form of counselling, which can be expanded to include psychological services for HIV clients who also present with PND. However, such women must first be subjected to routine screening as they attend postnatal services so that appropriate referrals are done.

The results of this study have linked PND as an outcome variable not only to socio-economic factors, but also to the clinical and obstetric factors.

The findings of this study have therefore expanded the understanding of PND in a predominantly rural setting for this population. The findings support the principle of integrating individual screening for PND as the associated factors vary with individuals. Such integrations will enable the diagnosing and treatment of PND and thus improve long term health outcomes, including the HIV disease progression which is worsened by undiagnosed and untreated PND.

## Figures and Tables

**Table 1 ijerph-17-08425-t001:** Sociodemographic, clinical, baby and obstetric characteristics of the sample (*n* = 386).

Characteristics		Value *n* (%)
Sociodemographic Information		
Maternal Age M (SD)		29 (5.8)
Marital status	Married	56 (14.5)
	Single/never married	330 (85.5)
Place of residence	Rural	316 (81.9)
	Town	70 (18.1)
Religion	Christianity	278 (72)
	Traditional	98 (25.4)
	Other	10 (2.6)
Education level	Completed matric	276 (71.5)
	No matric	110 (28.5)
Employment status	Not employed	304 (78.8)
	Employed	82 (21.2)
Monthly income	<R2 000	120 (64.9)
	R2 001-R5000	40 (21.6)
	>R5 000	40 (13.5)
Income source	Child support grant	183 (53)
	Family support	162 (47)
Parity	Uniparous	114 (29.53)
	Multiparity	272 (70.5)
Mode of delivery	Normal vaginal	255 (66.1)
	C-section	131 (33.9)
Place of delivery *	Hospital	251 (65.3)
	Clinic	124 (32.3)
	Home	9 (2.3)
Premature birth	Yes	67 (17.4)
	No	319 (82.6)
Baby’s sex	Girl	194 (50.3)
	Boy	192 (49.7)
Preferred sex	Yes	205 (53.1)
	No	128 (33.2)
	No preference	53 (13.7)
Planned pregnancy **	Yes	124 (32.7)
	No	255 (67.3)
Baby hospitalized **	Yes	59 (15.7)
	No	317 (84.3)
Infant feeding method	Breastfeeding	222 (57.5)
	Formula	155 (40.2)
	Both	9 (2.3)
Baby health status	Good health	356 (92.2)
	Ill health	30 (7.8)
HIV diagnosis *	Within one year	189 (49.2)
	More than a year	195 (50.8)
Partner’s HIV status	Positive	188 (50.4)
	Negative	60 (16.1)
	Don’t know	125 (33.5)
Family know HIV status	Yes	308 (80.2)
	No	76 (19.8)
HIV diagnosis during pregnancy *	Yes	223 (58.5%)
	No	158 (41.5)
Know baby’s status *	Yes	217 (56.5)
	No	167 (43.5)

* missing values > 10, ** Missing values < 10.

**Table 2 ijerph-17-08425-t002:** Bivariate analysis of maternal variables on PND symptoms.

Characteristics		Not Depressed EPDS ≤ 11 *n* = 222 (%)	Depressed EPDS ≥ 12 *n* = 164 (%)	*p*-Value	OR (95% CI)
Age	<25 years	67 (56.3)	52 (43.7)	0.98	1.0 (0.73–1.2)
	26–35 years	129 (58.1)	90 (41.1)		
	>35 years	26 (54.2)	22 (45.8)		
Marital status	Married	35 (62.5)	21 (37.5)	0.42	0.8 (0.42–1.41)
	Not married	187 (56.7)	143 (43.3)		
Place of residence	Rural	187 (59.2)	129 (40.8)	0.16	1.4 (0.86–2.44
	Town	35 (50)	35 (50)		
Religion	Christianity	152 (54.2)	126 (45.3)	0.15	0.7 (0.41–1.12)
	Traditional	65 (66.3)	33 (33.7)		
	Other	5 (50)	5 (50)		
Level of education	Matric	172 (57.5)	104 (37.7)	0.003	0.5 (0.32–89))
	No matric	50 (45.5)	60 (54.6)		
Employment status	Not employed	174 (57.2)	130 (42.8)	0.833	0.9 (0.58–1.55)
	Employed	48 (58.5)	34 (41.5)		
Monthly income	<R2000	100 (82)	22 (18)	0.001	2.3 (1.45–3.54)
	R2001–R5000	26 (65)	14 (35)		
	>R5000	11 (47.8)	12 (52.2)		
Income source	Child support grant	114 (62.3)	69 (37.7)	0.05	1.5 (1.1–2.36)
	Family support	84(51.8)	78(48.1)		
Parity	Uniparous	67 (57.6)	45 (42.4)	0.99	1.0(0.64–1.57)
	Multiparity	161 (57.5)	119 (42.5)		
Time of HIV diagnosis	One year	75 (55.6)	60 (44.4)	0.56	0.9 (0.59–1.35)
	≥2 years	146 (58.6)	103 (41.2)		
Partner HIV+	Yes	106 (56.4)	82 (43.6)	0.15	1.2 (0.92–1.89)
	No	42 (70)	18 (30)		
Family know HIV status	Yes	179 (58.1)	129 (41.9)	0.65	1.1 (0.78–1.96)
	No	42 (55.3)	34 (44.7)		
HIV diagnosis during pregnancy	Yes	133 (59.6)	90 (40.4)	0.31	0.8 (0.52–1.25)
	No	86 (54.4)	72 (45.6)		
Know baby HIV status	Yes	114 (52.5)	103 (47.5)	0.047	0.7 (0.58–0.99)
	No/awaiting results	108 (64.7)	59 (35.3)		

**Table 3 ijerph-17-08425-t003:** Bivariate analysis of baby and obstetric variables on PND symptoms.

Characteristics		Not Depressed EPDS ≤ 11 *n* = 222 (%)	Depressed EPDS ≥ 12 *n* = 164(%)	*p*-Value	OR (95% CI)
Baby age	0–4 weeks	67 (52.8)	60 (47.2)	0.32	0.98 (0.78–1.13)
	5–8 weeks	89 (60.5)	58 (39.5)		
	9–12 weeks	66 (60)	46 (41.1)		
Mode of delivery	Normal vaginal	150 (58.8)	105 (41.2)	0.47	1.2 (0.75–1.81)
	C-Section	72 (55.6)	59 (45)		
Place of birth	Hospital	141 (56.2)	110 (43.8)	0.80	0.9 (0.61–1.41)
	Clinic	77 (62.1)	47 (37.9)		
	Home	3 (33.3)	6 (66.7)		
Preterm delivery	No	201 (62.4)	121 (37.6)	0.000	3.4 (1.86–6.32)
	Yes	21 (32.8)	43 (67.2)		
Baby’s gender	Girl	111 (57.2)	83 (42.8)	0.90	0.9 (0.63,1.49)
	Boy	111 (57.8)	81 (42.2)		
Preferred baby gender	Yes	124 (60.5)	81 (39.5)	0.89	1.0 (0.75–1.49)
	No	71 (55.5)	57 (44.5)		
	No preference	27 (50.1)	26 (49.1)		
Pregnancy planned **	Yes	75 (60.2)	49 (39.5)	0.38	1.2 (0.88–1.98)
	No	142 (55.7)	113 (44.3)		
Baby hospitalized **	Yes	21 (35.6)	38 (64.4)	0.000	2.9 (1.62–5.26)
	No	195 (61.5)	122 (38.5)		
Feeding method	Breastfeeding	131(59)	91(42)	0.30	1.2 (0.84–1.87)
	Formula	88 (56.8)	67 (43.2)		
	Both	3(33.3)	6(67.7)		
Baby health status	Good health	214 (60.1)	142 (39.9)	0.000	4.1 (1.71–11.03)
	Ill health	8 (26.7)	22 (73.3)		

** Missing values < 10.

**Table 4 ijerph-17-08425-t004:** Multivariate analysis of selected variables on PND symptoms.

PND	Odds Ratio	*p* Value	95% Conf. Interval
Level of education	0.90	0.582	0.63–1.30
Income	2.25	0.015	1.17–4.35
Source of income	2.81	0.023	1.15–6.88
Know baby HIV status	0.47	0.065	0.21–1.05
Prematurity	1.93	0.331	0.51–7.37
Baby’s health	0.61	0.760	0.03–13.69

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
