# Peer review of "Clinical and Obstetric Risk Factors for Postnatal Depression in HIV Positive Women: A Cross Sectional Study in Health Facilities in Rural KwaZulu-Natal"

_ijerph, 2020, doi:10.3390/ijerph17228425_

Round 1

Reviewer 1 Report

This is a clearly written, interesting paper on an important topic,  It is straight forward.  The conclusions are drawn from the data. It would be helpful if the authors make a few suggestions how psychological support for PND might be provided  in areas where the incidence is high and the provision of health services is frequently stretched.

 I think that the formatting of the paper can be improved.  It seems that the tables can be presented in a more concise form since most of the data presented in the tables are stated in the text.

Author Response

Find attached the responses to the comments raised during the review process

Regards

Reviewer 2 Report

Mbatha et al. studied the clinical and obstetric risk factors for postnatal depression in HIV positive women. This study is well-designed and reveals some interesting findings. Their data support the conclusion pretty well. My only have some concerns on the format and the way how they present different statistical parameters.

  1. Abstract. I don’t think they need to present all those OR, CI and p values in abstract. They should use some more descriptive sentences to explain their findings but not the numbers. In addition, they also forgot to spell out OR, CI.

  1. There should be spaces between ‘=’ and numbers, like p-value = 0.05.

  1. The p-values should be displayed in same decimal spaces (Other values too). By the way, the author used ‘p’ at some places and used ‘p-value’ and ‘P value’ in other places. They should be consistent.

  1. How should a p value be zero?

Author Response

Find attached the responses to the review comments

Regards

Reviewer 3 Report

Thank you for the opportunity to read your work. The study concerns a very important topic of postnatal depression PND. Especially since it includes a group of women diagnosed with HIV. The work is written in a consistent and transparent manner. However, I have two observations:
1. Authors should be careful in describing the PND and the results. The EPDS score only allows you to determine the risk of PND and thus should appear in the work (which is indicated in the limitation section).
2. Of course, the EPDS can be used on the second day after delivery, but maybe it would be better to exclude women with children under 4 weeks of age. The  study group would be more uniform and this would be the time when PND is diagnosed (which will not arise during pregnancy).
3.  verse 157 unnecessary space

Author Response

Find attached responses regarding the matters raised in the review process

Regards
